# Automatic Wake and Deep-Sleep Stage Classification Based on Wigner–Ville Distribution Using a Single Electroencephalogram Signal

**DOI:** 10.3390/diagnostics14060580

**Published:** 2024-03-08

**Authors:** Po-Liang Yeh, Murat Ozgoren, Hsiao-Ling Chen, Yun-Hong Chiang, Jie-Ling Lee, Yi-Chen Chiang, Rayleigh Ping-Ying Chiang

**Affiliations:** 1Department of Intelligent Technology and Application, Hungkuang University, Taichung 433, Taiwan; d88543003@ntu.edu.tw; 2Asia-Pacific Branch, Innovative Medical and Health Technology Center (IMHTC), Taipei 114, Taiwan; murat.ozgoren@neu.edu.tr (M.O.); shirlin1971@gmail.com (H.-L.C.); jacqueline891008@gmail.com (J.-L.L.);; 3International Sleep Science and Technology Association (ISSTA), Taiwan Chapter, Taipei 104, Taiwan; 4Department of Biophysics, Faculty of Medicine and Department of Neuroscience, Brain and Conscious States Research Center, Near East University, Nicosia 99138, Cyprus; 5International Sleep Science and Technology Association (ISSTA), Headquarter, 10117 Berlin, Germany; 6Sleep Well International Chain Sleep Clinics, Taipei 114, Taiwan; 7Department of Executive Master of Business Administration, College of Management, National Taiwan Normal University, Taipei 106, Taiwan; 8Department of Entomology, College of Bioresources and Agriculture, National Taiwan University, Taipei 106, Taiwan; 9School of Medicine, Tzu-Chi University, Hua-Liang 970, Taiwan; 10Department of Otolaryngology Head and Neck Surgery, School of Medicine, China Medical University, Taichung 404, Taiwan; 11Department of Health Policy and Management, Bloomberg School of Public Health, Johns-Hopkins University, Baltimore, MD 21205, USA

**Keywords:** sleep EEG, sleep stage, automatic identification, time-frequency analysis, Wigner-Ville distribution, particle swarm optimization

## Abstract

This research paper outlines a method for automatically classifying wakefulness and deep sleep stage (N3) based on the American Academy of Sleep Medicine (AASM) standards. The study employed a single-channel EEG signal, leveraging the Wigner–Ville Distribution (WVD) for time–frequency analysis to determine EEG energy per second in specific frequency bands (δ, θ, α, and entire band). Particle Swarm Optimization (PSO) was used to optimize thresholds for distinguishing between wakefulness and stage N3. This process aims to mimic a sleep technician’s visual scoring but in an automated fashion, with features and thresholds extracted to classify epochs into correct sleep stages. The study’s methodology was validated using overnight PSG recordings from 20 subjects, which were evaluated by a technician. The PSG setup followed the 10–20 standard system with varying sampling rates from different hospitals. Two baselines, T1 for the wake stage and T2 for the N3 stage, were calculated using PSO to ascertain the best thresholds, which were then used to classify EEG epochs. The results showed high sensitivity, accuracy, and kappa coefficient, indicating the effectiveness of the classification algorithm. They suggest that the proposed method can reliably determine sleep stages, being aligned closely with the AASM standards and offering an intuitive approach. The paper highlights the strengths of the proposed method over traditional classifiers and expresses the intentions to extend the algorithm to classify all sleep stages in the future.

## 1. Introduction

Sleep is a vital and natural process that is integral to the maintenance of both physical and mental health in humans. It is reported that about 33% of the general population experience sleep problems [1]. Sleep disorders can have serious implications, potentially causing mood disorders such as depression or anxiety, reduced concentration or reaction time, decreased decision-making or learning capabilities, or even death. 

The diagnostic process for sleep problems begins with obtaining the patient’s all-night polysomnography (PSG) while they sleep. The PSG includes an electroencephalograph (EEG), electrooculogram (EOG), electrocardiogram (ECG), and chin electromyogram (EMG) among others. A sleep technician then visually scores the PSG, classifying each 30 s data epoch into different sleep stages. These findings are represented in hypnograms, providing physicians with a broad view of the sleep structure. 

The Rechtschaffen and Kales (R&K) [2] and the American Academy of Sleep Medicine (AASM) [3] are the two widely used sleep scoring systems. The R&K’s standard, established over forty years ago, divides sleep into seven stages: wake, non-rapid eye movement (NREM) sleep stages 1 to 4, rapid eye movement (REM) sleep, and body movement. The AASM refined the R&K standard by removing the body movement stage and categorizing it under the sleep or wake stages. Further, stages 3 and 4 from the R&K rules were merged to form slow wave sleep (SWS), also known as N3. As such, the AASM standard divides sleep into five stages: wake, NREM1 to NREM3 (N1 to N3), and REM sleep (termed as “stage R”).

Sleep scoring is a time-intensive procedure that may produce different results due to the potential subjectivity of the sleep technician. As a result, automatic sleep staging is deemed necessary for time efficiency and objective sleep stage assessments. 

Automatic sleep staging relies on methods such as rule-based methods, numerical classification methods, and hybrid systems [4]. Rule-based methods use signal patterns and human knowledge to establish scoring rules, but integrating data patterns and human knowledge can be challenging [5]. Numerical classification methods, popular in the last decade, typically extract PSG features through spectrum analysis or time-frequency domain analysis [6,7] and input these into a classifier that does not require human knowledge or signal patterns. Common classifiers include artificial neural networks, support vector machines, K-nearest neighbor (KNN), fuzzy classifiers, and hidden Markov models [8,9,10]. Feature selection is critical, and some studies have included large numbers of features in their studies, reaching up to 129 [11]. However, the operation of black-box-like classifiers may not be comprehensible to a human operator. While hybrid systems combine the advantages of both methods, they can be more complex to implement, with their efficacy often resting on the rule-based methods [12].

Previous research has utilized multi-channel signals to devise automatic sleep staging systems, finding that the increased number of recordings can potentially improve system accuracy [13,14,15,16]. These studies have employed between 4 and 48 PSG recordings, yielding high-performance results, with accuracies ranging from 50% to 85%. However, these techniques are not conducive to in-home systems. As an alternative, scoring methods based on single channels have been developed, providing satisfactory results [17,18,19,20,21,22,23]. Yet, due to the limited amount of information these methods provide, developing an automatic scoring system using a single channel requires more effort. 

This study seeks to algorithmically mirror the scoring process of sleep technicians based on the AASM standard, specifically for the wake and deep sleep (N3) stages. For this purpose, the authors consulted sleep technicians over the course of a year to ensure that the features extracted in this study adhered to both the AASM rules and the technicians’ interpretation. 

Firstly, the EEG signal was divided into epochs, with the Wigner–Ville Distribution (WVD) utilized to calculate the EEG energy per second across different bands. The WVD was chosen as it conveniently determines the energy per second and frequency bands through a time–frequency analysis. 

Subsequently, the energy of the whole band was calculated to identify major body movements. The frequency components of the bands were extracted to feature the wake and N3 stags. Furthermore, Particle Swarm Optimization (PSO) was adopted to pinpoint the optimal thresholds for the wake and N3 stages. While PSO has been previously used in various applications, it has not been applied in the optimization of scoring the sleep stage until now. Lastly, the features of each epoch, combined with the optimal thresholds, were used to classify each epoch as either the wake or N3 stage.

## 2. Material

In this study, twenty overnight PSG recordings were evaluated to classify the stages of wakefulness and N3 sleep. The recordings were provided from three different hospitals in Taiwan, namely, Taipei Chang Gung Memorial Hospital, Shin Kong Wu Ho-Su Memorial Hospital, and Kang-Ning General Hospital, contributing 4, 9, and 7 recordings, respectively. A single technician scored all twenty recordings. 

The twenty subjects of the study comprised seven males and thirteen females aged between 18 and 60 years. The average age was 34.1 years, with a standard deviation of 11.7 years. Before the study, all subjects underwent a medical history interview to assess their sleep quality. None of the subjects had pre-existing conditions such as cardiovascular diseases, epilepsy, or apnea. Each subject gave informed consent for the sleep study procedures.

In terms of the PSG measurement setup, the electrode placement followed the internal 10–20 standard system. Recordings from the three hospitals were performed at different sampling rates: Taipei Chang Gung Memorial Hospital recorded at 512 Hz, Shin Kong Wu Ho-Su Memorial Hospital at 256 Hz, and Kang-Ning General Hospital at 200 Hz. 

The initial segmentation of the PSG recordings was into 30 s periods or epochs. A qualified sleep technician then visually scored these epochs based on the AASM rules, classifying each into stages of wakefulness, NREM (N1, N2, N3), or REM sleep. The technician’s scoring was later used as the gold standard. 

The PSG recordings encompassed six EEG channels: C3, C4, M1, M2, M4, O1, and O2. For this study’s focus on classifying the wakefulness and N3 sleep sages, the C3M2 channel was employed. This channel was frequently used by sleep technicians for visual scoring, whilst the other EEG channels served as auxiliary. To demonstrate the universal applicability and feasibility of the proposed methods, half of the subjects were used as training data to obtain two thresholds while the remaining half were used to validate the performance of the proposed algorithm.

## 3. Methods

Figure 1 outlines a sleep staging protocol based on the AASM (American Academy of Sleep Medicine) standards. The focus of this study is on classifying wakefulness (stage wake) and slow wave sleep (stage N3), represented by the dotted line in Figure 1. Specific frequency bands, determined by both the AASM standards and expert opinion, were used as criteria. These include the following: δ: [0.5 4.0] Hz, slow wave for N3 stage.θ: [4.0 8.0] Hz, for wake stage.α: [8.0 13.0] Hz, for wake stage.T: [0.3 35] Hz, the recorded filter settings of AASM, for major body movement.

During the staging protocol, PSG (polysomnography) recordings are first segmented into epochs. The energy per second of each epoch is then calculated to identify significant body movement. The need to identify major movements comes from the AASM standards, which removes sleep time associated with such movements. AASM defines significant movement as instances where EEG readings are so obscured that the sleep stage is indeterminable.

Once an epoch is identified as having significant body movement, it is classified based on the presence of specific frequencies: If α rhythm, associated with wakefulness, appears at any point during the epoch, it is scored as “awake”.If there is no discernable α rhythm, but an epoch that could be scored as “awake” precedes or follows one with significant body movement, that epoch is also scored as “awake”.Otherwise, the epoch is scored in the same stage as the following epoch.

Those epochs not linked to significant body movement are classified based on the presence of specific frequencies associated with wakefulness or the N3 stage.

Figure 2 explains the implementation process using three steps: 1. pre-processing, 2. feature extraction, and 3. classification. The flowchart in Figure 2 displays the proposed sleep staging method, with details provided below.

### 3.1. Preprocessing

The C3M4 channel was preprocessed using the Wigner–Ville distribution, a form of time–frequency analysis (TFA). The *WVD* is a time–frequency analysis that exhibits the energy density of a signal in the time–frequency domain. The TFA was developed to deal with the drawbacks inherent in the Fourier transform (FT) since the FT theory is built on the assumption of a stationary signal [24], However, the EEG is a non-stationary signal with frequency content that varies with time.

For a time signal x(t), the *WVD* of x(t) is defined as follows:(1)$$WVD_{x}(t,f)=\int_{-\infty }^{\infty }x(t+\frac{\tau }{2})x^{*}(t-\frac{\tau }{2})e^{-j2\pi f\tau }d\tau$$
where * represents the complex conjugate. The $WVD_{x}(t,f)$ is a real-valued function because the product of $x(t+\frac{\tau }{2})x^{*}(t-\frac{\tau }{2})$ possesses Hermitian symmetry in τ. 

There are other branches of Fourier-based TFA such as the short-time Fourier transform (STFT) and the wavelet transform (WT). These methods basically perform Fourier transform for signals in a moving window. Usually, the application of Fourier-based TFA is limited by the uncertainty principle. 

Unlike the Fourier-based TFA, *WVD* is not restricted by the uncertainty principle. Hence, the optimal time and frequency resolutions can be achieved simultaneously [25]. WVD also preserves the frequency marginal condition for any signal, i.e., the marginal spectrum of *WVD* is equal to the power spectrum of the signal if the signal length is infinite: (2)$$\int_{-\infty }^{\infty }WVD_{x}(t,f)dt={\left|X(f)\right|}^{2}$$

Thus, one can use the marginal spectrum of *WVD* to determine the frequency distribution of the brain wave precisely. The aforementioned properties suggest that *WVD* could be an ideal tool for TFA. However, *WVD* has a critical drawback: cross-term. Since the cross-term hinders the application of *WVD*, previous research focused on how to remove the cross-terms. Most of them used a low-pass window function to filter the cross-term. However, the removal of the cross-term is achieved by sacrificing the time-frequency resolution. As Qian [26] indicated, some specific window functions reduce the frequency resolution of *WVD* equivalent to that of STFT or WT. However, it can be demonstrated that the cross-terms automatically vanish in the marginal spectrum. As a result, one can directly integrate the original signal to derive a marginal spectrum free from any contamination by the cross-terms. 

### 3.2. Feature Extraction

In this study, we analyzed the energy per second and overall energy throughout the night for specific bandwidths [δ, θ, α, T] as indicators for distinguishing the wake and N3 stages. To achieve this, we utilized the *WVD* spectrogram, integrating along each second to compute the energy per second for the pre-selected bandwidths. These were labeled as $E_{\delta }^{s}$, $E_{\theta }^{s}$, $E_{\alpha }^{s}$, and $E_{T}^{s}$, respectively. Also, the cumulative night energy for the specific bandwidths was determined by integrating the *WVD* spectrogram for the entire time axis and denoted as $E_{\delta }^{all}$, $E_{\theta }^{all}$, $E_{\alpha }^{all}$, and $E_{T}^{all}$, respectively. 

The AASM standard distinguishes large body movements as those where movement and muscle artifacts mask the EEG for more than half the epoch or generate indecipherable sleep stages. Refer to Figure 3a to see the wave pattern associated with significant body movement. Around the 17th second of this epoch, a high-amplitude wave appears, often induced by electrode movement resulting from physical movement. An epoch exhibiting high-amplitude oscillation for most of its duration is typically assessed by sleep technicians as a wake stage. Here, the rationale is that high-amplitude EEG signals indicate high energy. Consequently, the total energy of EEG, $E_{T}^{s}$, and $E_{T}^{all}$ are used as feature indicators of significant body movement. 

The AASM criteria categorize the wake stage as containing an α rhythm for more than 15 s of an epoch [3]. Refer to Figure 3b to view the EEG for wake stage. It shows dense sinusoidal waves superimposed on low-frequency waves. Sleep technicians tend to focus on the dense sinusoidal waves when interpreting sleep scores, often overlooking the low-frequency wave patterns within these scores. It is noteworthy that the α rhythm was detected through a sinusoidal pattern rather than by counting exact α rhythm periods. Hence, the α rhythm energy $E_{\alpha }^{all}$ was incorporated as an identifier of the wake stage.

We discovered that in most wake stage episodes, the α band energy was higher than the θ band energy; however, the opposite was not always the case. Figure 4a shows the *WVD* marginal spectrum of Figure 3b. From the marginal spectrum of the wake stage, one can see that a rough hump appears in the range of the α band, and the energy of the θ band is lower than that of the α band. Accordingly, the θ band energies $E_{\theta }^{s}$ and $E_{\theta }^{all}$ were included as features of the wake stage.

It is evident in Figure 4a that the highest peak, the low-frequency wave, occurs at 0.4 Hz, close to the δ band. This can be attributed to body movements or baseline wander and so on. In practice, sleep technicians generally disregard the δ band when identifying the α rhythm.

Stage N3 is identified as having 20% or more of epochs consisting of slow wave activity. According to the AASM standard, slow wave activity is defined as “a wave of frequency 0.5 Hz to 2 Hz and peak-to-peak amplitude greater than 75 μV, measured across the frontal regions.” [3]. The EEG of stage N3, as shown in Figure 3c, has a longer period than that of the wake stage. Similar to other stages, sleep technicians classify the epoch as stage N3 by identifying the slow wave pattern.

The *WVD* spectrum of Figure 3c can be seen in Figure 4b. The *WVD* spectrum shows a crest around the δ band, indicating that the dominant frequency in stage N3 is the slow wave. Consequently, the $E_{\delta }^{s}$ of each epoch was utilized to identify stage N3.

### 3.3. Classification

This study categorizes sleep stages following the AASM standard, which is notably different from the majority of automatic sleep scoring systems which primarily use classifiers such as artificial networks or support vector mechanics [27,28,29]. Conventional classifiers often resemble a “black box”, with their internal operations obscured and, therefore, difficult to understand. Additionally, these classifiers generally require more mathematical parameters to improve accuracy, which may be complex to comprehend. Thus, this study emphasizes the development of a technician-like classification algorithm to make stage scoring more intuitive and easily understandable.

#### 3.3.1. Major Body Movement

As depicted in Figure 1, the initial step in sleep staging is identifying major body movements. Since the existence of high-amplitude EEG signals is an indicator of body movement, circumstances (1) $E_{T}^{s}$ exceeding the default threshold and (2) the duration of each epoch are used to identify these movements. Given individual variations, this process is normalized using the total nighttime EEG energy, as outlined below:(3)$$P_{M}=\frac{E_{T}^{s}}{E_{T}^{all}}$$
where $P_{M}$ is the normalized value of $E_{T}^{s}$. A threshold $T_{M}=5\times 10^{-4}$ was set after analyzing all the data in this study. For each second, as $P_{M}>T_{M}$, this second of an epoch is identified as the major body movement. Another condition is that the body movement may sometimes lose the electrodes and result in no signal in practice. Therefore, while (1) $P_{M}>T_{M}$ or (2) $P_{M}=0$ is more than 15 s of an epoch, this epoch is identified as a major body movement according to the AASM rules. 

#### 3.3.2. Wake Stage

The sleep technician visually scores the wake stage by identifying the α rhythm and ignoring the θ rhythm. Hence, the energy difference $E_{\alpha }^{s}-E_{\theta }^{s}$ is extracted as the feature of the wake stage. Considering the individual difference again, the energy difference is normalized using the energy of the all-night θ rhythm, $E_{\theta }^{all}$, as follows: (4)$$P_{W}=\frac{E_{\alpha }^{s}-E_{\theta }^{s}}{E_{\theta }^{all}}$$

A threshold $T_{1}$ is also needed as the baseline of $P_{W}$. When $P_{W}>T_{1}$ and there is more than 15 s for an epoch, this epoch is classified as the wake stage. It should be noted that the threshold $T_{1}$ may not be easily selected as in the major body movement. The optimization of the threshold is utilized in the following section. 

Considering the right branch of Figure 1, once the epoch is classified as a major body movement, the next step is to determine whether any α rhythm exists. When an α rhythm exists, even if it is less than 15 s, this epoch is scored as the wake stage. Conversely, if an α rhythm does not exist, the preceding or following epoch is wake, and this epoch is still scored as the wake stage. 

#### 3.3.3. Stage N3

When an epoch consists of 20% or more of slow wave activity, this epoch is scored as stage N3 by the sleep technician. Considering the percent constituent and magnitude of slow wave, this paper uses the 80th percentile of $E_{\delta }^{s}$ to classify the sleep stage N3 of the corresponding epoch. The 80th percentile means the value below 80% of the observations. Suppose there are *M* observations that are sorted from least to greatest. The $P^{th}$ percentile is defined as follows:(5)$$n=\frac{P}{100}\times M$$

For example, in this study, the first step of the percentile is to sort the $E_{\delta }^{s}$ of an epoch in ascending order. Then, set *M* = 30 (the second of an epoch), so that *n* = (80/100) × 30 = 24. The 80th percentile is the 24th value of the sorted observations, i.e., the 6th maximum $E_{\delta }^{s}$ of an epoch. 

Again, considering the individual difference, the 80th percentile of $E_{\delta }^{s}$ is normalized by $E_{\delta }^{all}$ as follows: (6)$$P_{N3}=\frac{E_{\delta }^{s}}{E_{\delta }^{all}}$$

In the scoring process, the 80th percentile of $P_{N3}$ in each epoch is selected. Also, a threshold $T_{2}$ is set as the energy baseline of $P_{N3}$. For each epoch, when $P_{N3}>T_{2}$, this epoch is scored as stage N3.

#### 3.3.4. Optimization

Two energy baselines, denoted as $T_{1}$ and $T_{2}$, are required in identifying the stages of wake and N3. Determining these thresholds manually can prove challenging and time-consuming due to the lack of a presupposed range. Therefore, selecting the optimal thresholds to improve scoring accuracy becomes an intricate task. 

To solve this problem, we employed Particle Swarm Optimization (PSO) in this study to discover the optimal thresholds. This technique was initially introduced by Kennedy and Eberhart [30], and it is inspired by the group behavior observed in birds while foraging. Each particle seeks its best solution while communicating with others, simulating social interactions rather than individual cognitive abilities.

As compared to other optimization techniques, the PSO algorithm is uncomplicated and provides a quicker search speed [31]. Unlike traditional optimization methods including gradient descent and quasi-newton methods which optimize problems using a gradient, PSO can solve optimization problems that are not differentiable. Furthermore, PSO does not present the “black box” nature of optimization, such as artificial neural networks, boasting a limited ability to explicitly identify potential causal relationships [32].

In the PSO algorithm, each particle i is composed of three vectors: ${\vec{x}}_{i}$ is the current position, ${\vec{p}}_{i}$ is the personal best position, and ${\vec{v}}_{i}$ is the velocity. Moreover, the global best position ${\vec{p}}_{g}$ and the objective function (fitness function) f(x) are necessary for the algorithm. The current position ${\vec{x}}_{i}$ is considered a solution in the search space on each iteration of the algorithm. If the current position ${\vec{x}}_{i}$ is better than the historical position that has been searched so far, the ${\vec{x}}_{i}$ coordinates are stored in the vector ${\vec{p}}_{i}$. Similar to the swarm, the global best result so far is stored in ${\vec{p}}_{g}$. On the iterations step for particle i, the algorithm adjusts ${\vec{v}}_{i}$ and then adds ${\vec{v}}_{i}$ to ${\vec{x}}_{i}$ to determine the new position. Each particle keeps finding better positions and updating ${\vec{p}}_{i}$ and ${\vec{p}}_{g}$ until the criterion is achieved. 

The procedure of PSO is summarized as follows [33]:Initialization: set the random positions and velocities to all particles in the search space.Evaluate the fitness function $f(x_{i})$ for each particle i.If $f(x_{i})>f(p_{i})$, set ${\vec{x}}_{i}$ → ${\vec{p}}_{i}$. Determine ${\vec{p}}_{g}=\max[p_{i}]$ and update ${\vec{v}}_{i}$ and ${\vec{x}}_{i}$ according to the following equations:(7)$$\left\{\begin{array}{l} {\vec{v}}_{i}\leftarrow \omega \cdot {\vec{v}}_{i}+c_{1}\cdot r_{1}\cdot (p_{i}-x_{i})+c_{2}\cdot r_{2}\cdot (p_{g}-x_{i})\\ x_{i}\leftarrow x_{i}+v_{i} \end{array}\right.$$If the criterion is not met, go to step 2.

In Equation (7), ω is the inertia weight. $r_{1}$ and $r_{2}$ are random numbers. $c_{1}$ and $c_{2}$ are the correction factors. Poli et al. [33] described the effectiveness of these parameters in detail. This study set ω as the random number, $c_{1}=c_{2}=1$, the swarm size = 200, and the criterion is that iterations = 40. 

### 3.4. Performance Evaluation 

Sleep staging frequently uses measures such as classification accuracy, sensitivity, and the coefficient κ to evaluate the performance of proposed methods. These measures are derived from a confusion matrix. The confusion matrix displays the overlap of actual and predicted instances; in our case, the classification is into two stages labeled “yes” and “no”. The gold standard for comparison is the scoring results given by a sleep technician.

In the matrix, TP and TN refer to the number of epochs accurately classified as “yes” and “no”, respectively. FP denotes the number of epochs inaccurately classified as “yes” when they were actually “no”; likewise, FN denotes the number of epochs inaccurately classified as “no” when they were actually “yes”.

These values from the matrix allow us to calculate sensitivity (otherwise termed the TP rate) and accuracy with the following formulas:Sensitivity = TP/(TP + FN);Accuracy = (TP + TN)/(TP + TN + FP + FN).

Sensitivity, also known as true positive rate or recall, measures the proportion of actual positive cases correctly identified by a model. A high sensitivity indicates that the model is effective at capturing and correctly identifying positive cases, minimizing false negatives.

Accuracy measures the overall correctness of the model by calculating the ratio of correctly predicted instances to the total instances. Accuracy provides a general assessment of the model’s correctness across all classes. However, it may be influenced by class imbalances.

Sensitivity focuses on the model’s ability to correctly identify positive instances, while accuracy provides an overall measure of correctness across all classes. It is crucial to consider both metrics in tandem to gain a comprehensive understanding of a model’s performance, especially in scenarios where the cost of false positives or false negatives may vary. High sensitivity and accuracy values indicate superior classification results, demonstrating the efficacy and reliability of the proposed algorithm.

Another important performance measure is Cohen’s kappa coefficient. This coefficient assesses the level of agreement between separate raters—in this instance, the stage scoring performed by the proposed method and the sleep technician [34]. A high κ indicates that the system is in good agreement with the human technician’s scoring and that its decisions are reliable. It is particularly robust as it accounts for an agreement that may have occurred due to chance, making it a more reliable measure than simple per cent agreement calculations.

Kappa is defined as follows:(8)$$\kappa =\frac{p_{o}-p_{c}}{1-p_{c}}$$
where $p_{o}$ is the proportion of agreement and $p_{c}$ is the proportion of agreement expected by chance. From Table 1, $p_{o}$ and $p_{c}$ can be calculated using the following equations:(9)$$p_{o}=\frac{A+D}{N}$$
(10)$$p_{c}=\frac{1}{N}\times [(\frac{N1\times N3}{N})+(\frac{N2\times N4}{N})]$$
(11)N=A+B+C+D

Landis and Koch [35] interpreted κ as follows: less than 0.00 is poor agreement; 0.00 to 0.20 is slight agreement; 0.21 to 0.40 is fair agreement; 0.41 to 0.60 is moderate agreement; 0.61 to 0.8 is substantial agreement; more than 0.80 is excellent agreement. However, this guideline is not universally accepted. Fless [36] characterized the agreement as follows: over 0.75 as excellent; 0.75 to 0.4 as fair to good; below 0.40 as poor. 

In Section 3.3.4, a fitness function is required for PSO. In this study, we chose the classification accuracy as a fitness function. The reasoning behind this choice is that a higher True Positive rate could result in the proposed algorithm scoring excessively and, consequently, increasing the error rate.

## 4. Results and Discussion 

This study focuses on the classifying of the wake and deep sleep (N3) stages. Hence, in the following discussion, the sleep stages are categorized as the wake stage, N3 stage, and other stages. 

Ten subjects were randomly chosen as the training set. The features, $P_{W}$ and $P_{N3}$, of the 10 EEG were input into the PSO algorithm to find the optimal thresholds $T_{1}$ and $T_{2}$. This study found the optimal values as $T_{1}=1.61\times 10^{-6}$ and $T_{2}=5.87\times 10^{-5}$. Then, for the other 10 subjects, treated as the test set (i.e., 50:50 partition between training and test data), the two thresholds were used to classify the sleep stages. These 10 subjects had a total of 8207 epochs. The technician’s scoring results were as follows: wake stage: 2107 epochs, stage N3: 1412 epochs, and other stages: 4688 epochs. 

Table 2 shows the confusion matrix of scoring the wake stage between the proposed algorithm and the sleep technician. The sleep stages are categorized into wake and others. The category “others” contains sleep stages N1, N2, N3, and REM. The sensitivity and accuracy of the agreement on the wake stage are 93.55% and 92.04%. In addition, the kappa coefficient κ is 0.80. The results show that the parameter $P_{W}$ can be a good feature for the sleep and wake stages. 

Table 3 is the confusion matrix of stage N3. The sensitivity, accuracy, and κ of stage N3 are 89.45%, 92.62%, and 0.76, respectively. The results also show that $P_{N3}$ can be used as a feature for sleep stage N3. 

Table 4 shows the confusion matrix of global performance. Here, the sleep stages are categorized into wake, N3, and others (including N1, N2, and REM). The global accuracy of agreement is 85.46%. 

Khalighi [15] presented the performance evaluation of recent works on automatic sleep stage classification. These researchers used multiple features (5 to 129 features) and multiple channels (10 to 47 recordings). The sensitivity of the wake stage was from 34.00% to 88.59%, and the sensitivity of stage N3 was from 82.00% to 92.90%. In addition, the global accuracy of the agreement was from 55% to 80%. The sensitivity of stages wake and N3 in Khalighi’s research was 88.59% and 87.13%, respectively. Compared to the most important published works, our proposed method used fewer features with a single channel. Moreover, the proposed classification directly followed the AASM standards. The classified procedure is easy to comprehend. 

Accuracy and sensitivity are evaluated as follows:Accuracy: Table 2 shows an accuracy of 92.04% for the wake stage, and Table 3 shows 92.62% for stage N3, indicating that the scoring system closely matches the technician’s scoring for these stages.Sensitivity: Table 3 shows that the sensitivity for N3 is 89.45%, meaning the system is quite good at correctly identifying the N3 sleep stage when it is present.

Accuracy and sensitivity can help clinicians and researchers judge how well the automatic scoring system performs against a human scorer and are crucial for ensuring that patients receive accurate assessments of their sleep patterns. As our system achieved high accuracy and high sensitivity, it can be considered reliable for use in both clinical and research settings.

The features of the subject with better results are illustrated to show the classification effect of the proposed method. The all-night EEG contains a total of 736 epochs. The technician’s scoring results are as follows: wake stage: 115 epochs; stage N3: 249 epochs. Figure 5d is an example of the hypnogram that was scored by the sleep technician. 

Figure 5a shows another example of the normalized total energy $P_{M}$ all night. The horizontal line in Figure 5a is the threshold line of major body movement, $T_{M}=5\times 10^{-4}$. The major body movement appears in epoch #174~#187.

Figure 5b shows bar graphs of $P_{W}>T_{2}=1.61\times 10^{-6}$ for an existing time that is more than 15 s of an epoch. The *Y*-axis is limited to 15–30 s. Comparing Figure 5b with Figure 5d, one can see that the tendency of the bar graph is similar to the hypnogram in the wake stage. $P_{W}>T_{2}$ contains more than 15 s mainly in epochs #1~#27, #64~#72, #114~#118, #135~#168, and #560~#580. This shows that the proposed method is effective in identifying the wake stage.

Figure 5c shows the all-night 80th percentile of $P_{N3}$. The horizontal line in Figure 5c is the threshold $T_{2}=5.87\times 10^{-5}$. In Figure 5c, the humps appear in epoch #52~#57, #84~#113, #235~#262, #283~#360, #373~#450, and #710~#735. The tendency is also similar to the hypnogram in stage N3. This also shows that the 80th percentile of $P_{N3}$ is also an effective marker of sleep stage N3.

Figure 6 shows one of the worst results. Figure 6b shows the bar graphs of the all-night $P_{W}$. As in Figure 5b, each bar represents $P_{W}>T_{2}=1.61\times 10^{-6}$ and the existing time of more than 15 s of an epoch. The proposed method scored the wake stage approximately in four clusters: epoch #20~#67, #148~#257, #342~#427, and #519~#618. The technician scored the wake stage in epoch #1~#12. Comparing Figure 6b to Figure 6d, one can find that the epochs of the wake stage overlap stage N3 in epochs #20~#67, #148~#257, and #342~#427. Once the epoch is scored as the wake stage, this epoch will not be scored further in the proposed method. 

As for the all-night 80th percentile of $P_{N3}$ that is shown in Figure 6c, the amplitude of $P_{N3}$ is almost lower than the threshold $T_{2}$. Although the curve hump has the same tendency as that of stage N3 in epochs #25~#92, #158~#261, and #360~#432, the hump amplitude is too low to be scored as stage N3. 

To identify the difference between the above two cases, the EEGs of the 20 subjects in stages wake and N3 were analyzed. Two worse scoring results were found almost in the same hospital. Being the worse results means that the sensitivity and accuracy were less than 40% and 70%, respectively. 

Figure 7 shows the EEG of the above two cases in the wake stage and the N3 stage. Figure 7a,b show the EEG of the wake stage and N3 stage with better results. Figure 7c shows the worst EEG of the wake stage and N3 stage. Comparing Figure 7b with Figure 7d, one can find the α rhythm riding on the low-frequency δ rhythm in stage N3. Then, after rechecking the PSG recordings, it is found that all-night EEGs are full of the α rhythm. The inferred reason may be that the electrodes were not placed well, which caused the noise. 

The high accuracy of the results indicates that this automatic system is an innovative approach to sleep stage classification. The system presented in the study classifies wakefulness and deep sleep stages (N3) based on EEG signals using the Wigner–Ville distribution, adhering to AASM standards and calibrated using technicians’ expertise. With the employment of Particle Swarm Optimization for optimal threshold determination, this system facilitates objective and efficient data analysis, offering considerable benefits over traditionally manual and subjective analyses by technicians. The benefits include increased objectivity and time efficiency due to the automation of sleep stage classification, which is traditionally resource-intensive and prone to subjectivity. Additionally, potential applications span clinical sleep studies, home-based sleep monitoring, and research on sleep disorders, offering a cost-effective and user-friendly alternative for sleep assessments. The system’s robustness, demonstrated through sensitive feature extraction and straightforward algorithmic implementation, holds promise for broadening access to sleep quality evaluation and could be important to improve patient care and further sleep science research.

Apart from the training and testing dataset being split evenly, the performance of this classification method was further validated through three permutations, with 30% for the training set and 70% for the test set. The obtained results are as follows:Classification of Wake vs. Other Stages (N1, N2, N3, and REM): An average accuracy of 91.95% was achieved.Classification of N3 vs. Other Stages (Wake, N1, N2, and Rem): An average accuracy of 88.34% was achieved.Classification of Wake, N3 vs. Light-Sleep stages (N1, N2, and Rem): An average accuracy of 84.31% was achieved.

Even though the average accuracy in a 70%/30% dataset distribution is slightly lower than the 50%/50% dataset distribution, the values are close when they are compared with each other. Yet, the method’s ability to maintain high accuracy levels indicates its efficiency and the potential for practical application in sleep studies, where data availability and distribution can significantly vary.

Moreover, these findings highlight the method’s adaptability and its potential ease of integration into clinical and research settings, where accurate and reliable sleep stage classification is important. It opens paths for further research, particularly in optimizing model training with varied dataset distributions to harness maximum predictive accuracy. These explorations contribute significantly to the broader goal of enhancing sleep quality assessment and the diagnosis and treatment of sleep-related disorders.

## 5. Conclusions

This paper presents a methodology for classifying wake and deep sleep (N3) stages based on the AASM standard. A single EEG is initially transformed into the time–frequency domain by using the *WVD*. Subsequently, the energy of each EEG epoch is calculated to identify major body movements and for further examination. The frequency components of the δ, θ, and α bands are extracted as scoring features for each sleep stage. A PSO approach is employed to determine the optimal thresholds for the wake and N3 stage features. Eventually, these features for each epoch are used to categorize the corresponding epoch into either the wake stage or stage N3. 

The methodology aims to imitate a sleep technician’s visual scoring process, directly referencing their observations and the AASM standard. Unlike many previous studies, which utilized a variety of mathematical parameters for feature extraction, this study used frequency components of the δ, θ, and α bands. Furthermore, it strictly adhered to the AASM standard, unlike many studies that have used more oblique classification models. As a result, this model aligns more closely with the sleep technician’s reasoning process rather than an engineering perspective. 

The results indicate that our feature extraction and classification method is effective for sleep stage scoring. Despite potential variation due to individual hospital recordings, the sensitivity, accuracy, and kappa coefficient of this study outperform many prior studies. This suggests that our proposed algorithm is sufficiently reliable. Future work will aim to complete the classification of all sleep stages using technician-based rules.

## Figures and Tables

**Figure 1 diagnostics-14-00580-f001:**
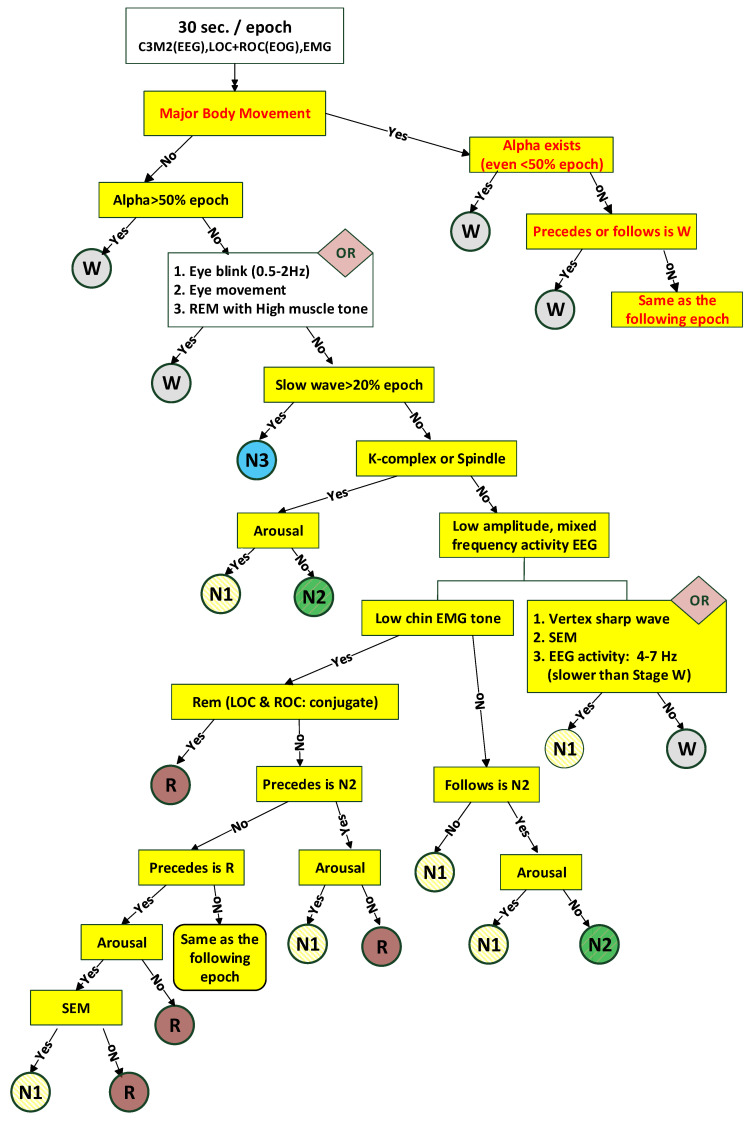
The procedure of sleep staging.

**Figure 2 diagnostics-14-00580-f002:**
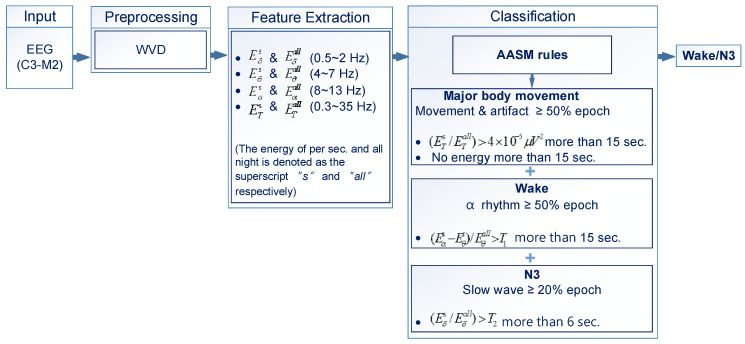
The proposed sleep staging method.

**Figure 3 diagnostics-14-00580-f003:**
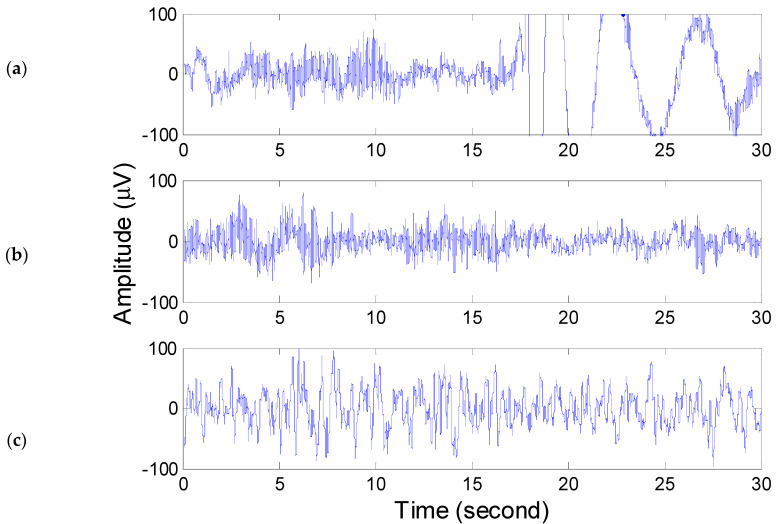
EEG of (**a**) major body movement, (**b**)wake, and (**c**) stage N3.

**Figure 4 diagnostics-14-00580-f004:**
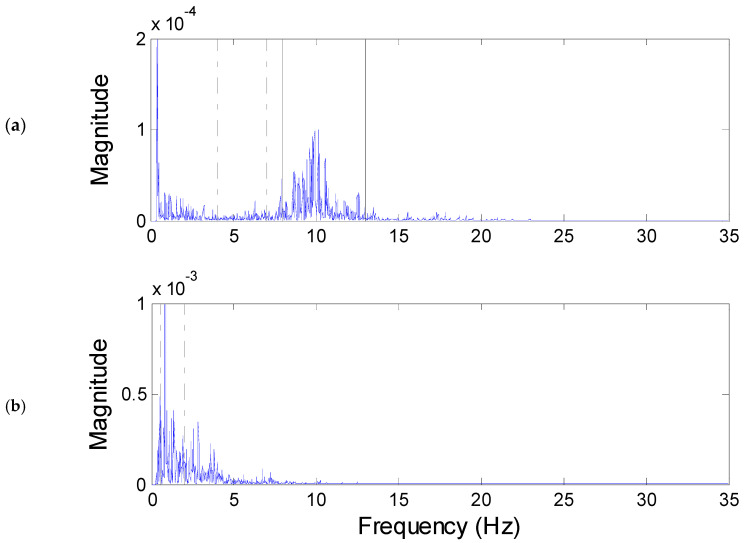
The *WVD* spectrum of EEG at (**a**) wake stage and (**b**) stage N3.

**Figure 5 diagnostics-14-00580-f005:**
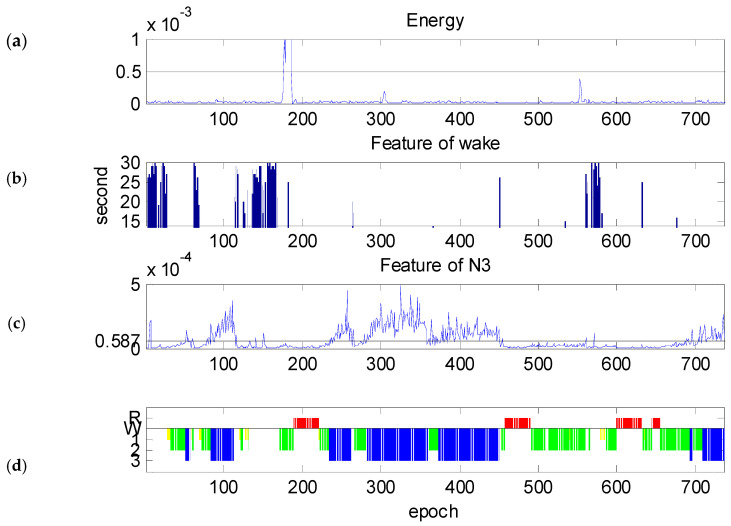
The features of one subject: (**a**) total energy $P_{M}$, (**b**) bar graphs of $P_{W}>T_{2}=1.61\times 10^{-6}$, (**c**) 80th percentile of $P_{N3}$, and (**d**) hypnogram where the horizontal line represents wake periods, yellow represents N1 stage, green represents N2 stage, blue represents N3 stage, and red represents REM.

**Figure 6 diagnostics-14-00580-f006:**
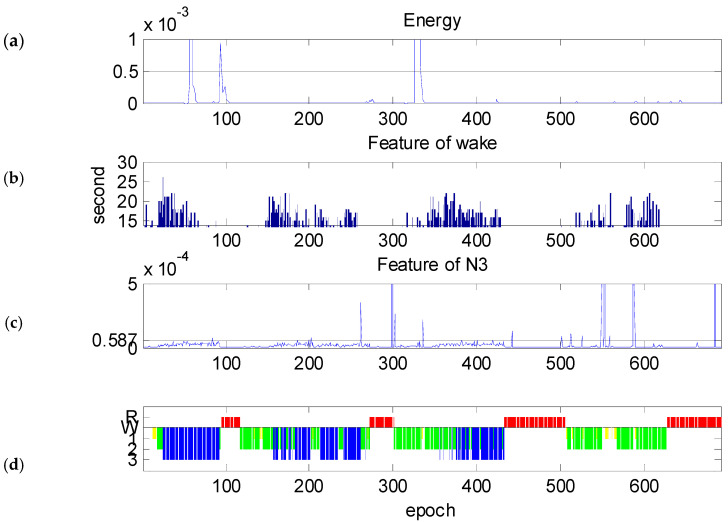
The features of one subject: (**a**) total energy $P_{M}$, (**b**) bar graphs of $P_{W}>T_{2}=1.61\times 10^{-6}$, (**c**) 80th percentile of $P_{N3}$, and (**d**) hypnogram where the horizontal line represents wake periods, yellow represents N1 stage, green represents N2 stage, blue represents N3 stage, and red represents REM.

**Figure 7 diagnostics-14-00580-f007:**
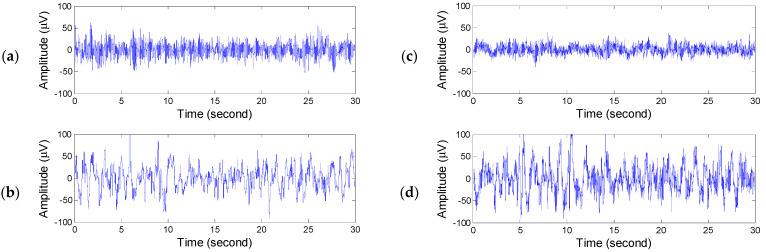
EEG of different individuals: (**a**) wake, (**b**) N3, (**c**) wake, (**d**) N3.

**Table 1 diagnostics-14-00580-t001:** Confusion matrix.

	Technician	Yes	No	
Automatic Scoring				
Yes		True Positive (TP): *A*	False Positive (FP): *B*	N3=A+B
No		False Negative (FN): *C*	True Negative (TN): *D*	N4=C+D
		N1=A+C	N2=B+D	

**Table 2 diagnostics-14-00580-t002:** Confusion matrix of wake stage.

		Technician’s Score Wake	(N1, N2, N3 & REM)
Automatic Score	Wake (N1, N2, N3 & REM)	1971 136	517 5583
		TP Rate (%) 93.55	TN Rate (%) 99.08
			Accuracy (%): 92.04

**Table 3 diagnostics-14-00580-t003:** Confusion matrix of stage N3.

		Technician’s Score N3	(Wake, N1, N2 & REM)
Automatic Score	N3 (Wake, N1, N2 & REM)	1263 149	457 6338
		TP Rate (%) 89.45	TN Rate (%) 93.27
			Accuracy (%): 92.62

**Table 4 diagnostics-14-00580-t004:** Confusion matrix of automatic scoring.

		Technician’s Score Wake	N3	(N1, N2 & REM)
Automatic Score	Wake N3 (N1, N2 & REM)	1971 36 100	30 1263 119	487 421 3780
		TP Rate (%) 93.55	89.45	80.63
				Accuracy (%): 85.46

## Data Availability

Restrictions apply to the datasets.

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
