# Peer review of "Automatic Wake and Deep-Sleep Stage Classification Based on Wigner–Ville Distribution Using a Single Electroencephalogram Signal"

_diagnostics, 2024, doi:10.3390/diagnostics14060580_

Round 1

Reviewer 1 Report

Comments and Suggestions for Authors

In this manuscript the authors report a novel classification approach to sleep stages. This can improve the time spent by a technician classifying the PSG data manually.

The authors clearly describe the state-of-the art and the goals of the work. I would suggest to improve the abstract has this is not representing the work as it is described in the text, consider rewriting it.

The methods are very well documented and the reader can follow, but I suggest to add a specific section regarding statistical analysis that include the tests used (might just be needed a rephrasing of the sections that describe the sensitivity and accuracy steps).

The results are well structured, however, I would like to see the results with the leave-one-out strategy or the test-retest 30%/70% of the sampling using permutations. E.g. it is not clear if the trainning set is always the same and the test too. To test different sub-sets can give different results? I suggest the authors add this and include the range of results accuracy in the section. I think if this maintains the results the work will be stronger.

As this is a key point I suggest the acceptance of the paper conditioned to see this work performed first.

Comments on the Quality of English Language

can be improved in some paragraphs.

Reviewer 2 Report

Comments and Suggestions for Authors

The manuscript presents an intriguing approach to scoring the stages of wakefulness and deep sleep (N3) using a combination of technician expertise and the AASM standard. The methodology involves the segmentation of EEG signals into epochs, followed by the application of the Wigner-Ville distribution (WVD) to quantify the EEG energy per second within specific frequency bands, including various components and the entire bandwidth. This approach is then leveraged to extract features indicative of major body movement, wakefulness (stage wake), and deep sleep (stage N3).One notable aspect of this study is the incorporation of particle swarm optimization (PSO) to determine optimal thresholds for identifying stage wake and stage N3. This demonstrates a commitment to precision and fine-tuning, which is critical in the field of sleep scoring.

However, current work could benefit from providing a bit more context about the significance of identifying these specific sleep stages, as well as any potential real-world applications or implications of the proposed method. Additionally, it would be helpful to mention the dataset used or the scale of the study to give readers a sense of the scope and generalizability of the findings.

Overall, the manuscript presents an interesting approach to sleep stage classification that combines expert knowledge with data-driven techniques. Further details and results in the full paper would be needed to assess the effectiveness and potential impact of this methodology in the field of sleep research and clinical practice, e.g., Figure 7.

Comments on the Quality of English Language

The overall English quality is quite poor, with numerous instances of awkward or unclear language that impedes comprehension.
